# Involvement of Glycogen Synthase Kinase 3β (GSK3β) in Formation of Phosphorylated Tau and Death of Retinal Ganglion Cells of Rats Caused by Optic Nerve Crush

Yurie Fukiyama [1], Takahisa Hirokawa [1], Shinji Takai [2], Teruyo Kida [1] and Hidehiro Oku [1,*]

[1] Department of Ophthalmology, Osaka Medical and Pharmaceutical University, Osaka 569-8686, Japan; mm080062@icloud.com (Y.F.); omc122@yahoo.co.jp (T.H.); teruyo.kida@ompu.ac.jp (T.K.)

[2] Department of Innovative Medicine, Osaka Medical and Pharmaceutical University, Osaka 569-8686, Japan; shinji.takai@ompu.ac.jp

[*] Correspondence: hidehiro.oku@ompu.ac.jp; Tel.: +81-726-83-1221

**Abstract:** Tauopathy is a neurodegenerative condition associated with oligomeric tau formation through abnormal phosphorylation. We previously showed that tauopathy is involved in death of retinal ganglion cells (RGCs) after optic nerve crush (ONC). It has been proposed that glycogen synthase kinase 3β (GSK3β) is involved in the hyperphosphorylation of tau in Alzheimer's disease. To determine the roles of GSK3β in tauopathy-related death of RGCs, lithium chloride (LiCl), a GSK3β inhibitor, was injected intravitreally just after ONC. The neuroprotective effects of LiCl were determined by counting Tuj-1-stained RGCs on day 7. Changes of phosphorylated (ser 396) tau in the retina were determined by Simple Western analysis (WES) on day 3. Retinal GSK3β levels were determined by immunohistochemistry (IHC) and an ELISA. There was a 1.9- and 2.1-fold increase in the levels of phosphorylated tau monomers and dimers on day 3 after ONC. LiCl significantly suppressed the increase in the levels of phosphorylated tau induced by ONC. GSK3β was mainly present in somas of RGCs, and ELISA showed that retinal levels increased to 2.0-fold on day 7. IHC showed that the GSK3β expression increased over time and remained in RGCs that were poorly stained by Tuj-1. The GSK3β and tau expression was colocalized in RGCs. The number of RGCs decreased from $1881 \pm 188$ (sham control) to $1150 \pm 192$ cells/mm$^2$ on day 7, and LiCl preserved the levels at $1548 \pm 173$ cells/mm$^2$. Accordingly, GSK3β may be a promising target for some optic nerve injuries.

**Keywords:** tauopathy; glycogen synthase kinase 3β (GSK3β); lithium chloride (LiCl); tau phosphorylation; optic nerve crush (ONC)

## 1. Introduction

Tau is a critical protein that plays a crucial role in maintaining the function of axons in the central nervous system (CNS). Tauopathies are neurodegenerative diseases that arise from the accumulation of toxic tau species in the CNS, and Alzheimer's disease is one such example. When axons are damaged due to various causes, tau becomes hyperphosphorylated and polymerized, leading to the formation of tau oligomers that have an intermediate size between the monomer and neurofibrillary tangles. Neurofibrillary tangles, which are aggregates of hyperphosphorylated tau protein, have long been considered a pathological biomarker of Alzheimer's disease. However, recent studies have indicated that neurofibrillary tangles themselves may not be toxic. Instead, tau oligomers, which are soluble and accumulate in nerve cell bodies or extracellular spaces, have toxic effects on the nervous system. One possible mechanism is impairment of axonal transport, leading to neuronal cell death. An increase in the expression of tau oligomers is also known to occur in the retina in animal models of Alzheimer's disease [1,2].

The optic nerve is included in the CNS and the pathological accumulation of tau occurs in the retinal ganglion cells (RGCs) after the optic nerve is injured. The reduction of tau accumulation by knockdown of the *tau* gene in experimental glaucoma has been shown to protect RGCs [3]. We have also shown that the knockdown of *tau* rescues RGCs from optic nerve crush (ONC) [4]. These findings suggest that tauopathy may occur in optic nerve pathology with a subacute or acute course of axonal injury, and that this mechanism is involved in the death of RGCs. Various optic nerve diseases are almost always accompanied by axonal injuries, leading to retrograde axonal degeneration and the death of RGCs. However, studies on tauopathy-associated optic nerve diseases remain scarce.

Excessive phosphorylation of tau is involved in the formation of tau oligomers. Based on pathological examinations of the brain in Alzheimer's disease, two proteins, cyclin-dependent kinase 5 (Cdk5) and glycogen synthase kinase 3β (GSK3β), may be involved in the abnormal phosphorylation of tau [5,6]. We have determined the effects of roscovitine, a selective inhibitor of Cdk5, and suggested that Cdk5 is associated with tau phosphorylation and the death of RGCs after ONC in rats [7]. The recent consensus is that Cdk5 and GSK3β are closely correlated in the formation of hyper-phosphorylated tau [8]. Cdk5 exaggerates the actions of GSK3β and these kinases cooperate in causing abnormal phosphorylation of tau [9]. Thus, it is vitally important to determine the roles of GSK3β in the phosphorylation of tau in the retina after ONC.

Tauopathy is well known to be associated with chronic neurodegeneration. Inactivation of *GSK3β* has a beneficial effect on axonal regeneration after ONC [10]. Thus, the purpose of this study was to assess whether GSK3β is involved in the death of retinal ganglion cells (RGCs) with an acute course after ONC in rats, through a mechanism of increased phosphorylated tau formation. To achieve this purpose, we examined changes in the expression of phosphorylated tau in the retinas of rats after ONC using immunoblotting. We also assessed changes in the expression of GSK3β and tau, as well as their localization in RGCs, by immunohistochemistry (IHC). Additionally, we determined the effect of lithium chloride (LiCl), a selective inhibitor of GSK3β, on phosphorylated tau levels using Simple Western analysis (WES). Furthermore, we evaluated the neuroprotective effect of the GSK3β inhibitor on RGCs by assessing their viability using an immune labeling technique.

## 2. Materials and Methods

### 2.1. Animals

Nine week old male Wistar rats were purchased from Japan SLC, located in Shizuoka, Japan. The rats were housed in a room maintained at a temperature of around 23 °C and a humidity level of 60%, with a light-dark cycle of 12 h each. All animal care and handling was conducted in adherence with the ARVO guidelines outlining the ethical use of animals in research related to ophthalmology and vision. The experimental procedures, which conformed to the Animal Research: Reporting In Vivo Experiments (ARRIVE) guidelines [11], were authorized by the Animal Use and Care Committee of Osaka Medical and Pharmaceutical University (Approval No. 21025-A). A total of 68 rats were used in this study.

### 2.2. Chemicals

Unless otherwise specified, all chemicals were obtained from Sigma-Aldrich Corp. (St. Louis, MO, USA).

### 2.3. Anesthesia and Euthanasia

General anesthesia was induced in all rats through an intraperitoneal injection of a combination of medetomidine (0.75 mg), midazolam hydrochloride (4.0 mg), and butorphanol tartrate (5.0 mg/kg body weight) during all surgeries [12]. To euthanize the rats, they were placed in a 13.8-L cage with wood-shaving bedding and exposed to $CO_2$ at a flow rate of 6 L/min [13]. After respiratory arrest, rats were placed in a small cage and

exposure to $CO_2$ was continued to confirm euthanasia until the animals were apparently unconscious with muscle relaxation, pupil dilatation, and pallor of retinal reflex.

### 2.4. Optic Nerve Crush

The animals were anesthetized, and a midline incision was made on the skull to expose the superior surface of the left eye. Incision of the superior rectus muscle exposed the left optic nerve, which was crushed 2 mm behind the eye using forceps for 10 s, with care taken to avoid occlusion of blood vessels and resultant retinal ischemia. Following the procedure, an ophthalmoscopic examination was conducted to verify the absence of retinal ischemia. We previously determined that our procedures did not cause retinal ischemia by confirming that the *HIF-1α* gene was unchanged on days 2 and 7 after ONC by real-time PCR [14]. In other animals, a sham operation was performed on the left eyes (sham controls), whereby the optic nerve was exposed in the same way, but not crushed. The right eyes of all animals were left untouched but were not used as controls because unilateral ONC may affect gene expression of the contralateral retina [15].

To determine the effects of the GSK3β inhibitor, lithium chloride (LiCl) was injected into the vitreous of rats just after ONC. The dose of 100 mM LiCl solution was 2.0 μL/eye; this was like the dose used to inhibit retinal GSK3β in rats [16]. The intravitreal injection of 2.0 μL of PBS was performed on placebo controls just after ONC.

### 2.5. Immunohistochemistry

Previous studies have shown that axotomy or ONC results in a delayed demise of RGCs, with the number of RGCs remaining unchanged for 5 days after the injury. Subsequently, the count decreases to 50% on day 7 and drops below 10% on day 14 [17]. Therefore, to determine the survival of RGCs, the evaluation was performed 7 days after ONC.

Following euthanasia, retinas were carefully extracted by removing the cornea and applying pressure with forceps from behind the eyeball. Subsequently, the isolated retinas were fixed in 4% PFA in PBS overnight and then washed with PBS. Subsequently, the retinas were blocked in PBS containing 1.0% bovine serum albumin (BSA) and 0.3% Triton X-100 before incubation with Alexa 488-conjugated mouse monoclonal neuron-specific class III β-tubulin (Tuj-1) antibody (1:500 dilution, Biolegend, San Diego, CA, USA). Tuj-1 is a marker for RGCs [18]. The retinas were left in the same medium overnight at 4 °C and then washed with PBS. Then, we carefully flat mounted the retinas on a glass slide with the vitreous side facing up, applied the sealing agent (Fluoromount/Plus; Diagnostic BioSystems, Pleasanton, CA, USA), and covered them with a cover slip the following morning.

To determine the number of surviving RGCs, the stained flat mounts were imaged with a fluorescence microscope (BZ X700; Keyence, Osaka, Japan). Four areas from the four quadrants of the retina, located at 1.5 mm from the edge of the optic disc, were photographed. Operating the ImageJ 1.51 software program (http://imagej.nih.gov/ij/ (accessed on 28 April 2018)); provided in the public domain by the National Institutes of Health, Bethesda, MD, USA), all cells with Tuj-1 staining in an area of $0.2 \times 0.2$ mm at the center of each image were counted (Figure 1).

The average cell density with Tuj-1-positivity per square millimeter was determined, and the decrease in RGCs was estimated by comparing the density in the experimental groups to that in the sham control group. A single observer (YF), blinded to whether the sample was from an experimental or sham animal, counted the number of Tuj-1-positive cells.

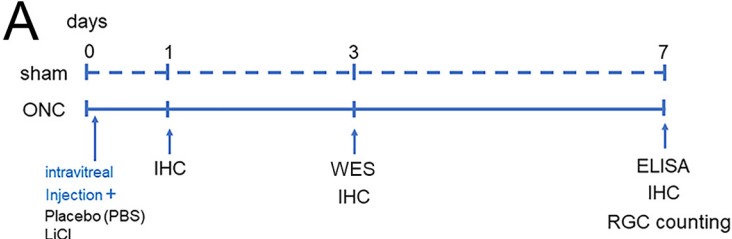

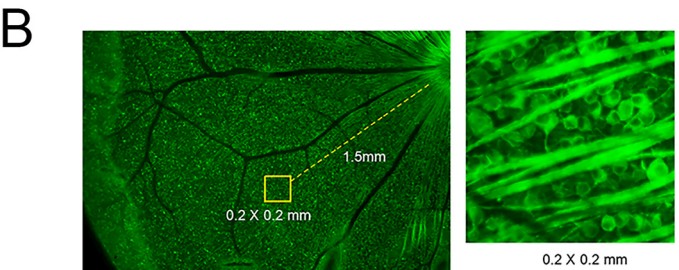

**Figure 1.** The experimental timeline is presented in (**A**). Animals were euthanized on days 0, 1, 3, and 7 after ONC to investigate temporal changes in the expression of GSK3β using IHC. Protein extraction was performed for WES on day 3 and for ELISA on day 7 (**A**). The protective effects on RGCs were determined on day 7, and the schematic diagram is shown in (**B**). Assays were conducted at 1.5 mm from the optic disc in each retinal quadrant ((**B**), left panel) and all Tuj-1-stained RGCs in an area of 0.2 × 0.2 mm were counted ((**B**), right panel).

In addition to IHC with Tuj-1 staining, retinas were incubated with Alexa 555-conjugated rabbit monoclonal antibody to tau (1:100, abcam). In the other IHC staining experiment, retinas were incubated with mouse monoclonal antibody to tau (1:100, abcam) and rabbit polyclonal antibody to GSK3β (1:100, abcam, Cambridge, MA, USA) overnight at 4 °C, and the localization of these proteins in the retina was determined. The confocal images of the retinas were acquired using a confocal laser microscope (TCS SP8, Leica, Wetzlar, Germany). In certain experiments, the fluorescence intensity of GSK3β in RGCs was assessed through IHC and semi-quantified using the ImageJ 1.51 software program to determine time-dependent changes in the expression.

### 2.6. Protein Simple Capillary Immunoassay (WES)

Three days after ONC, animals were euthanized, and their retinas were collected. This time point was chosen because the number of RGCs is stable, which is important to determine the causal relationship of RGC pathology. Each retina from every experimental animal was homogenized in RIPA buffer containing 50 mM Tris/HCl (pH 7.6), 150 mM NaCl, 5.0 mM KCl, 0.5% sodium deoxycholate, 1.0 mM EDTA, 1.0% nonidet P40, and 1.0% protease and phosphatase inhibitors (Merck, Darmstadt, Germany) in ice-cold water. After centrifugation at $10,000 \times g$ for 10 min at 4 °C, the supernatant was saved. The total protein concentration was measured by Qubit 4.0 Fluorometer (Thermo Fisher Scientific, Waltham, MA, USA).

Simple Western analysis (WES, ProteinSimple 004-600, Minneapolis, MN, USA) was performed to determine the changes in the levels of phosphorylated and total tau after ONC. This system enables us to detect various proteins sensitively with much less time than conventional immunoblot [19]. We focused on analyzing the phosphorylated tau at ser 396, which is one of the earliest sites of phosphorylation in Alzheimer's disease in the brain [20]. Phosphorylation at this site occurs in the retinas of rats with experimental glaucoma [3] and causes polymerization of tau [20]. In addition, it has been shown that LiCl inhibits tau phosphorylation at ser 396, leading to neuroprotection [21].

Briefly, the protein concentration extracted from each retinal sample was adjusted to 1.2 μg/μL in 4.0 μL of 0.1× WES sample buffer (ProteinSimple 042-195), and 1.0 μL of the fluorescent master mix (ProteinSimple PS-FL01-8) was added. The WES capillary plate was treated with EzBlock Chemi (ATTO, Tokyo, Japan), primary antibodies, appropriate secondary antibodies (Promega, Madison, WI, USA), and a luminol-peroxidase mix (15 μL). Rabbit monoclonal anti-phosphorylated tau at serine 396 (1:40, ab109390, Abcam, Cambridge, GBR) and mouse monoclonal anti-total tau (1:20, sc-390476, Santa Cruz, Dallas, TX, USA) were used as primary antibodies.

The WES system settings were as follows: sample loading (20 sec), blocking (30 min), incubations with primary antibodies (120 min), secondary antibodies (60 min), and luminol/peroxide chemiluminescence detection (high dynamic range: HDR).

The experimental groups consisted of three groups: sham control, placebo group (ONC + PBS intravitreal injection), and LiCl group (ONC + LiCl intravitreal injection). Assays were performed in biological replicates (*n* = 3).

### 2.7. ELISA

The retinal levels of glycogen synthase kinase-3β (GSK3β) were measured using sandwich ELISA kits (RayBiotech, Peachtree Corners, GA, USA) on day 7 after ONC, according to the manufacturer's protocols. After euthanizing the animals, retinas were isolated. After homogenizing the retina using T-PER in the presence of protease inhibitor (Merck, Darmstadt, Germany), the homogenized solution was centrifuged at $10,000 \times g$ for 10 min at 4 °C and the supernatant was collected. Each well in the 96 well-microplate was precoated with purified GSK3β antibodies. Every standard or sample solution was added into 2 wells to provide duplicate measurements in each sample. Samples were incubated overnight at 4 °C to form the GSK3β antibody-GSK3β complex. After washing, HRP-labeled GSK3β antibody was added into each well. Subsequently, TMB substrate was added to develop a blue color and the color intensity proportionally reflected the amount of GSK3β in each well based on the catalysis of the HRP enzyme activity. Finally, the developed color was converted into yellow using acid. Then, the absorbance (OD) at 450 nm was determined by a micro-plate reader (SH-100 Lab, Corona, Ibaraki, Japan). The GSK3β level was obtained by a calibration curve and standardized by protein concentration.

### 2.8. Statistical Analyses

The mean and standard deviation (SD) were used to express the data. The one-way ANOVA was used to compare means across groups, and if there was a significant difference, Scheffe post-hoc test was used to compare individual groups. The level of significance was set at $p < 0.05$.

### 2.9. Experimental Timeline

Figure 1 shows the time points for performing IHC, WES, and ELISA, as well as the method for counting retinal ganglion cells.

## 3. Results

### 3.1. Changes of Phosphorylated Tau after Optic Nerve Crush

On day 3 following ONC, we measured the levels of both total and phosphorylated tau by using WES assays. Figure 2 displays representative protein bands for both phosphorylated and total tau.

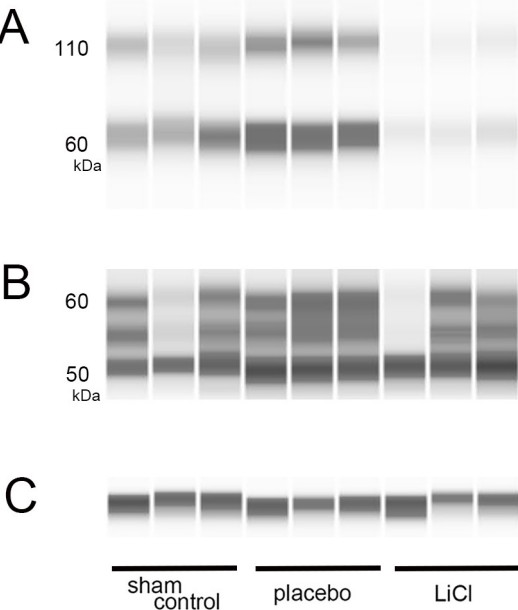

**Figure 2.** Levels of phosphorylated and total tau after ONC. (**A**–**C**): On day 3 following ONC, we performed WES assays to measure phosphorylated and total tau levels in the retina. (**A**) shows representative protein bands for phosphorylated tau at serine 396, while (**B**) displays bands for total tau. In (**C**), β-actin is shown as an internal control for the experiment. The band observed around 60 kDa in the assay of phosphorylated tau is likely the monomer (**A**). The bands observed around 110 kDa are considered to represent dimers, indicating that polymerization has occurred (**A**). The bands observed from 50- to 60-kDa are considered to represent total tau depending on the molecular weight predicted by the antibody used in the study (**B**).

Phosphorylation of tau at the serine 396 site can occur in the retina of a rat glaucoma model, resulting in increased protein bands at 50, 55, and 100 kDa in immunoblotting [3]. The antibody used in this study, abcam ab109390, was predicted to recognize a molecular weight of 60-kDa for phosphorylated tau at serine 396. We measured the levels of phosphorylated tau around the 60-kDa band (Figure 2A) and normalized them to the expression of β-actin, which served as an internal control (Figure 2C). In addition to the 60-kDa band, we observed bands around 110 kDa. The presence of these higher molecular weight bands suggests the possible polymerization of phosphorylated tau (Figure 2A). Thus, we also examined changes in the bands around 110 kDa, which may represent dimers of phosphorylated tau (Figure 2A).

Several isoforms of tau are known to exist, and the molecular weight predicted by the antibody to total tau (sc-390476, Santa Cruz) is 55-kDa. We detected presence of three bands within the range of 50 to 60 kDa, which may represent total tau. Subsequently, we quantified the total amounts of these bands in each group (Figure 2B).

Levels of phosphorylated and total tau following ONC were quantified and are presented in Figure 3 as fold changes compared to the sham control. The results demonstrated an increase in phosphorylated tau levels in the 60-kDa bands (Figure 3A, left panel), which may represent monomers, after ONC. In the placebo group, there was a significant $1.9 \pm 0.3$-fold increase in the level of phosphorylated tau monomers compared to the sham control ($p = 0.01$, Scheffe). LiCl treatment significantly reduced the increase by $0.3 \pm 0.1$-fold compared to the sham control level ($p < 0.01$, Scheffe) (Figure 3A, left panel).

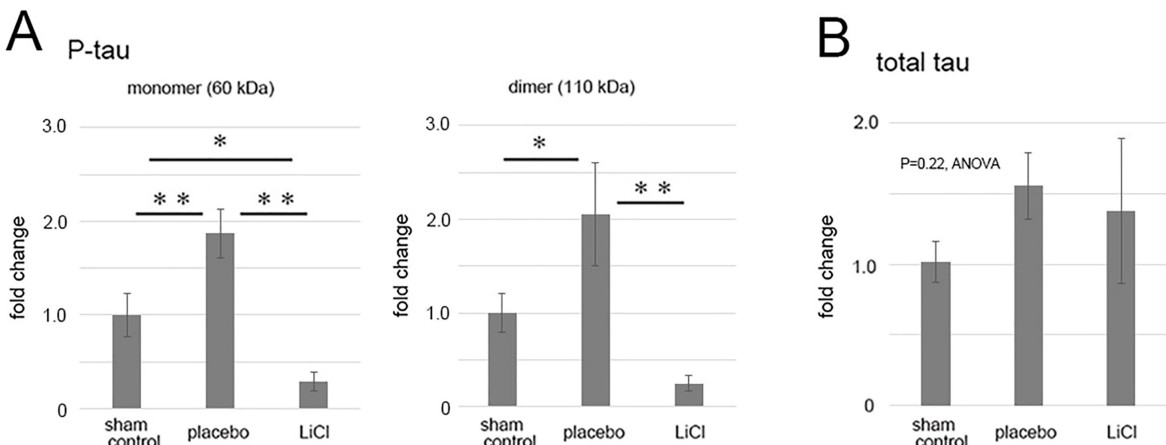

**Figure 3.** The figure illustrates the alterations in levels of phosphorylated and total tau after ONC. The results depict the fold-change (mean ± SD) of both monomer and dimer quantities of phosphorylated tau (**A**) and total tau (**B**) in comparison to the sham control. Following ONC, there was a notable increase observed in both the monomers (left) and dimers (right) of phosphorylated tau within the placebo group (**A**), indicating statistical significance. LiCl treatment exhibited a mitigating effect on the heightened levels of phosphorylated tau induced by ONC (**A**). Conversely, the levels of total tau remained relatively unchanged across the groups (**B**). The data was derived from three retinas representing three animals per group, with biological replicates ($n = 3$). * $p < 0.05$, ** $p < 0.01$, Scheffe.

In the placebo group, levels of dimers of phosphorylated tau also significantly increased to 2.1 ± 0.5-fold compared to the sham control ($p = 0.03$, Scheffe) (Figure 3A, right panel). LiCl treatment suppressed the increase by 0.3 ± 0.1-fold ($p < 0.01$, Scheffe) compared to the sham control level (Figure 3A, right panel).

Although there was a tendency of an increase in the levels of total tau after ONC, the changes in total tau levels among groups were not significant ($p = 0.22$, ANOVA, Figure 3B). In the placebo group, the total tau level was 1.56 ± 0.24-fold compared to the sham control level. Similarly, LiCl did not result in a decrease in the total tau level, and the level was 1.37 ± 0.51-fold from the sham control (Figure 3B).

*3.2. Effects of LiCl on Survival of RGCs after Optic Nerve Crush*

Figure 4 shows the effect of LiCl on RGC survival 7 days after ONC. The images were obtained from a flat-mounted retina using a confocal microscope. RGCs were double-stained with Tuj-1 (conjugated to Alexa 488) and tau (conjugated to Alexa 555). The number of cells immune-positive to Tuj-1, presumably RGCs, decreased after ONC, while immune-reactivity for tau was enhanced in the cell bodies of RGCs. In addition, speckled tau protein deposits were observed around the RGCs (Figure 4, placebo).

Decreased immune-reactivity to Tuj-1 was thought to occur in dying RGCs, and expression of tau appeared to be increased in these cells (Figure 4, placebo). These findings may suggest a causal relationship between increased tau expression and loss of RGCs. LiCl appeared to alleviate the degree of reduction in Tuj-1-positive cell numbers and the increase in immune reactivity to tau observed after ONC.

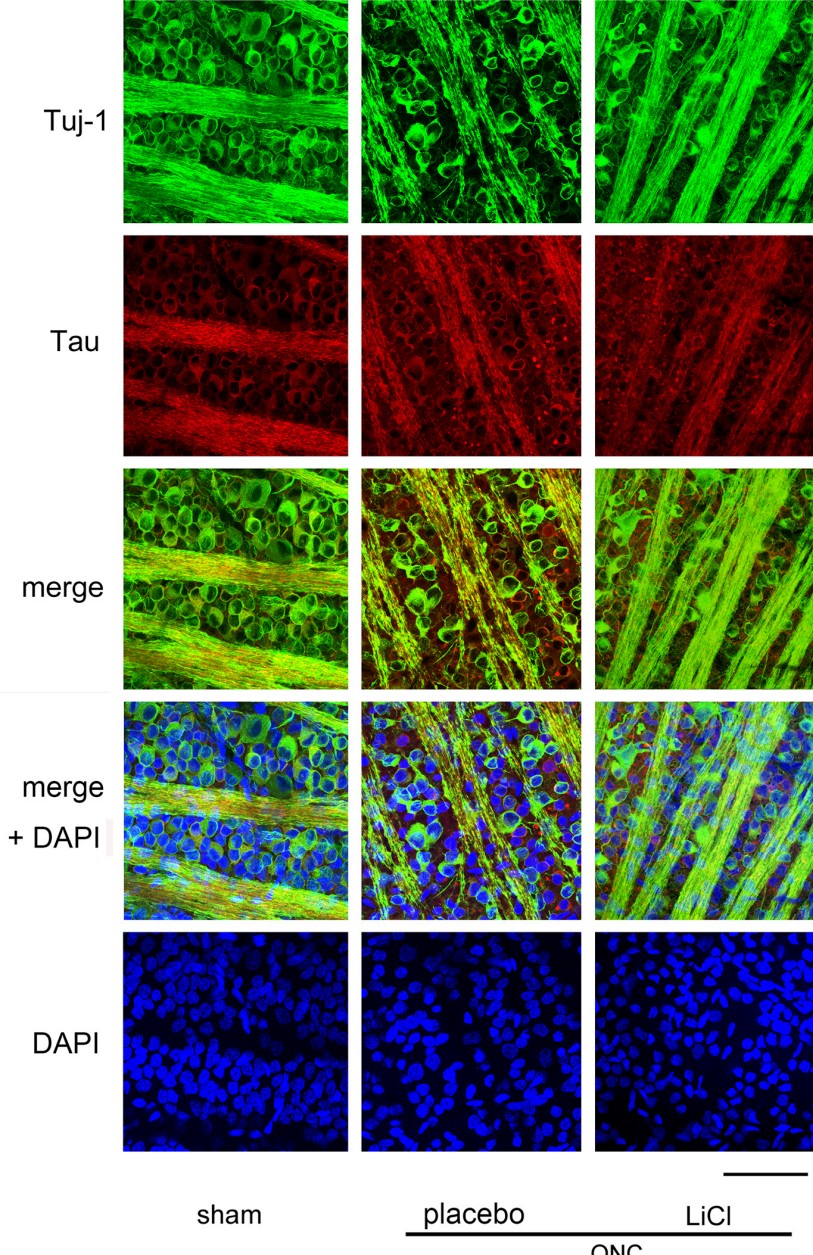

**Figure 4.** Representative samples of confocal microscopic images obtained from flat-mounted retinas, co-labelling with Tuj-1 and tau proteins. Samples were harvested on day 7 after ONC. In the sham control retina, cells stained with Tuj-1, presumably the RGCs (green) were thickly crowded, and tau (red) was primarily distributed in fine granules within the cell bodies of these cells. However, after ONC, there seemed to be a reduction in the number of RGCs with an increase in immune reactivity to tau. Speckled deposits of tau protein were observed surrounding the RGCs. The administration of LiCl appeared to mitigate the decrease in RGCs and the increase in immune reactivity to tau observed after ONC. Nuclei were stained with DAPI (blue). (Bar = 100 µm).

We then determined the number of surviving RGCs on day 7 using fluorescein microscope (BZ X700). Images were obtained approximately 1.5 mm from the margin of the optic disc. These images are demonstrated in Figure 5A as representative examples, and densities of surviving RGCs are shown in Figure 5B. In the sham control group ($n = 8$), the number of RGCs stained by the Tuj-1 antibody expressed by the mean $\pm$ SD was $1881 \pm 188$ cells/mm$^2$. On day 7 following ONC, the number significantly decreased to $1150 \pm 192$ cells/mm$^2$ in the placebo group ($n = 6$) ($p < 0.01$; Scheffe). The density of RGCs

was preserved at a significantly higher level of $1548 \pm 173$ cells/mm$^2$ in the LiCl group ($n = 8$, $p < 0.01$, Scheffe) on day 7.

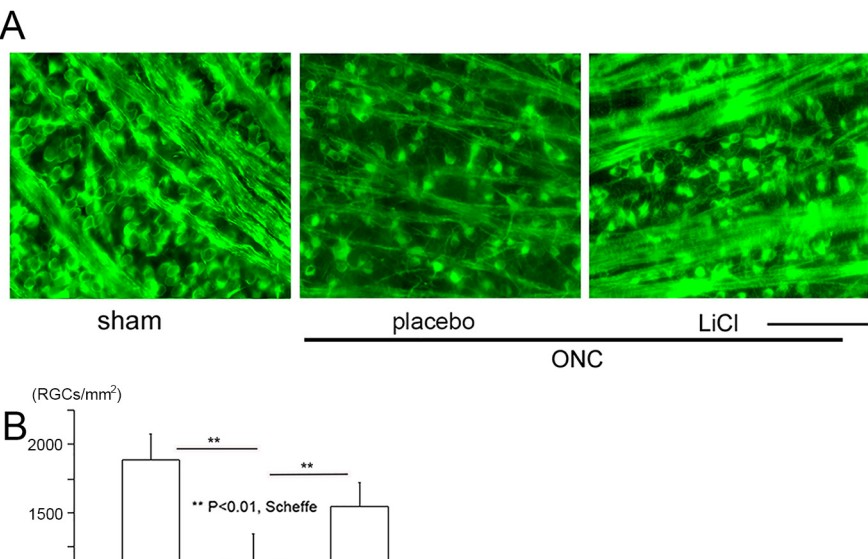

**Figure 5.** The protective effects of LiCl are demonstrated in representative flat-mount retinal images stained with Tuj-1. (**A**): The number of Tuj-1-positive RGCs was reduced after ONC (placebo), whereas treatment with LiCl prevented this reduction. (Bar = 100 μm). (**B**): Cell viability is expressed as the mean $\pm$ SD number of Tuj-1-stained RGCs per square millimeter (cells/mm$^2$). After ONC, the mean $\pm$ SD number of Tuj1-stained RGCs significantly decreased, while LiCl significantly reduced the degree of decline.

*3.3. Determination of the Expression of GSK3β by Immunohistochemistry*

Figure 6 displays characteristic confocal images of flat-mounted retinas, demonstrating temporal changes in GSK3β expression in RGCs, captured approximately 1.5 mm from the optic disc margin. Tuj-1-positive RGCs were highly dense in the sham control retinas and on days 1 and 3 post-ONC. However, on day 7, the number of Tuj-1-positive cells decreased significantly (as shown in Figure 6).

Immune reactivity to GSK3β was present in the somas of the RGCs of the sham control. The intensity of the immune reactivity to GSK3β appeared to become stronger on days 1, 3, and 7. The fluorescence intensity was quantified using the ImageJ 1.51 software program and stated in AIUs (arbitrary intensity units). The levels of fluorescence intensity in the sham control presented in the mean $\pm$ SD were $34.4 \pm 4.6$ AIU, and they increased to $38.3 \pm 5.6$, $41.2 \pm 5.5$, and $44.1 \pm 8.8$ AIU on days 1, 3, and 7, respectively, after ONC (Figure 6).

Another example showing changes in the expression of Tuj-1 and GSK3β in the retinal ganglion cells 7 days after ONC are presented in Figure 7. Immune reactivity to GSK3β was intensified after ONC and was still present in dying RGCs, where immune-reactivity to Tuj-1 was weak (Figure 7).

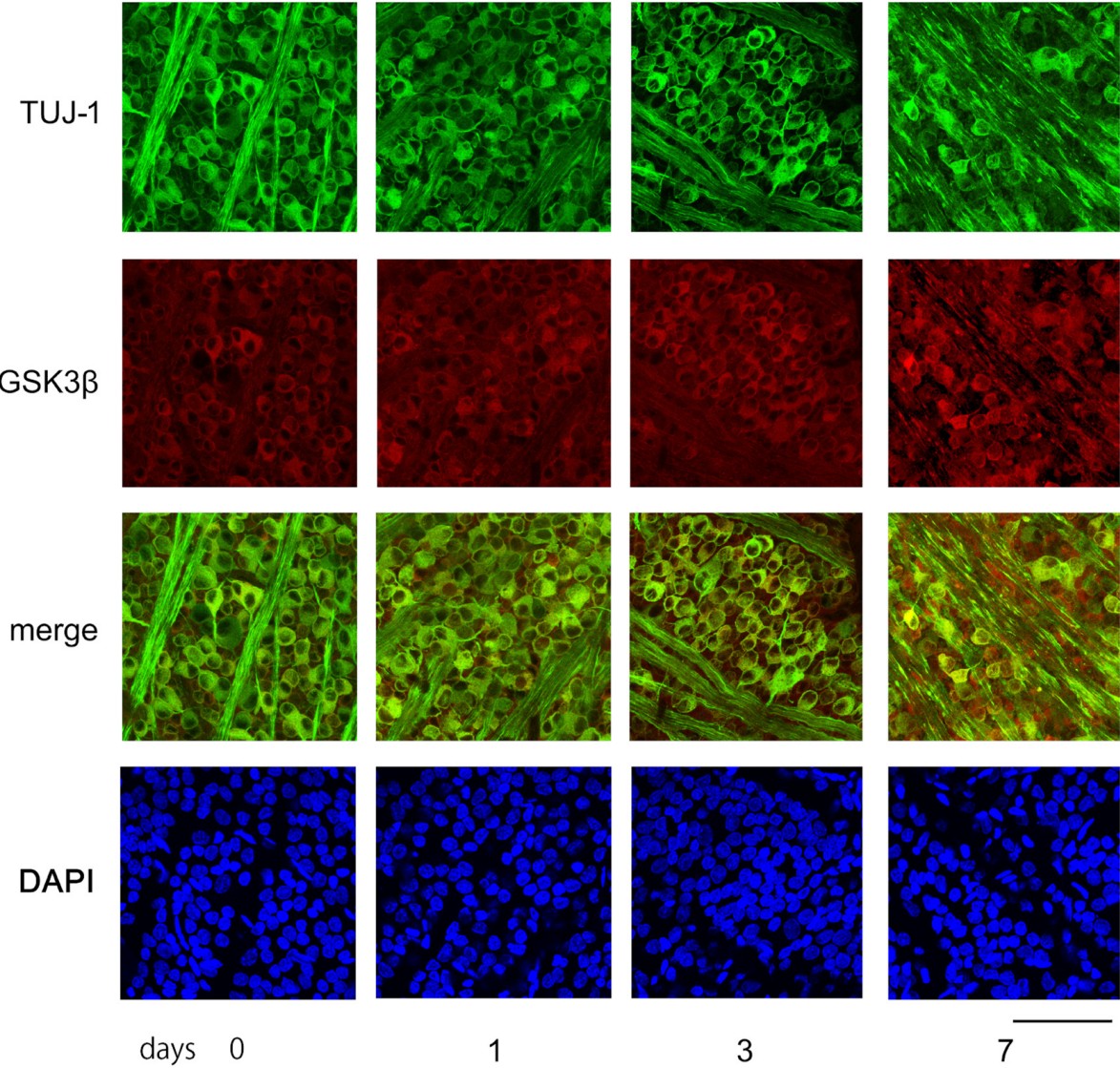

**Figure 6.** The figure depicts examples of confocal images obtained from flat-mounted retinas that illustrate changes over time in the levels of Tuj-1 and GSK3β proteins. In the control retina (day 0), Tuj-1-stained RGCs (green) were dense, while GSK3β (red) seemed to be mainly present in the somas of RGCs. The number of retinal ganglion cells (RGCs) with immune positivity to Tuj-1 remained constant on days 1 and 3, but the expression of GSK3β appeared to increase over time. On day 7 after ONC, the number of Tuj-1-stained cells clearly decreased with increased immune-reactivity to GSK3β. Nuclei were stained with DAPI (blue). (Bar = 100 μm).

We then performed IHC to determine whether the expression of GSK3β and tau was colocalized. As shown in Figure 8, GSK3β was present in the somas of the RGCs, and the expression of tau was almost even in the sham control (sham). In contrast, tau immune reactivity was intensified in some RGCs where the expression of GSK3β was also dense, suggesting the role of GSK3β in the expression of tau.

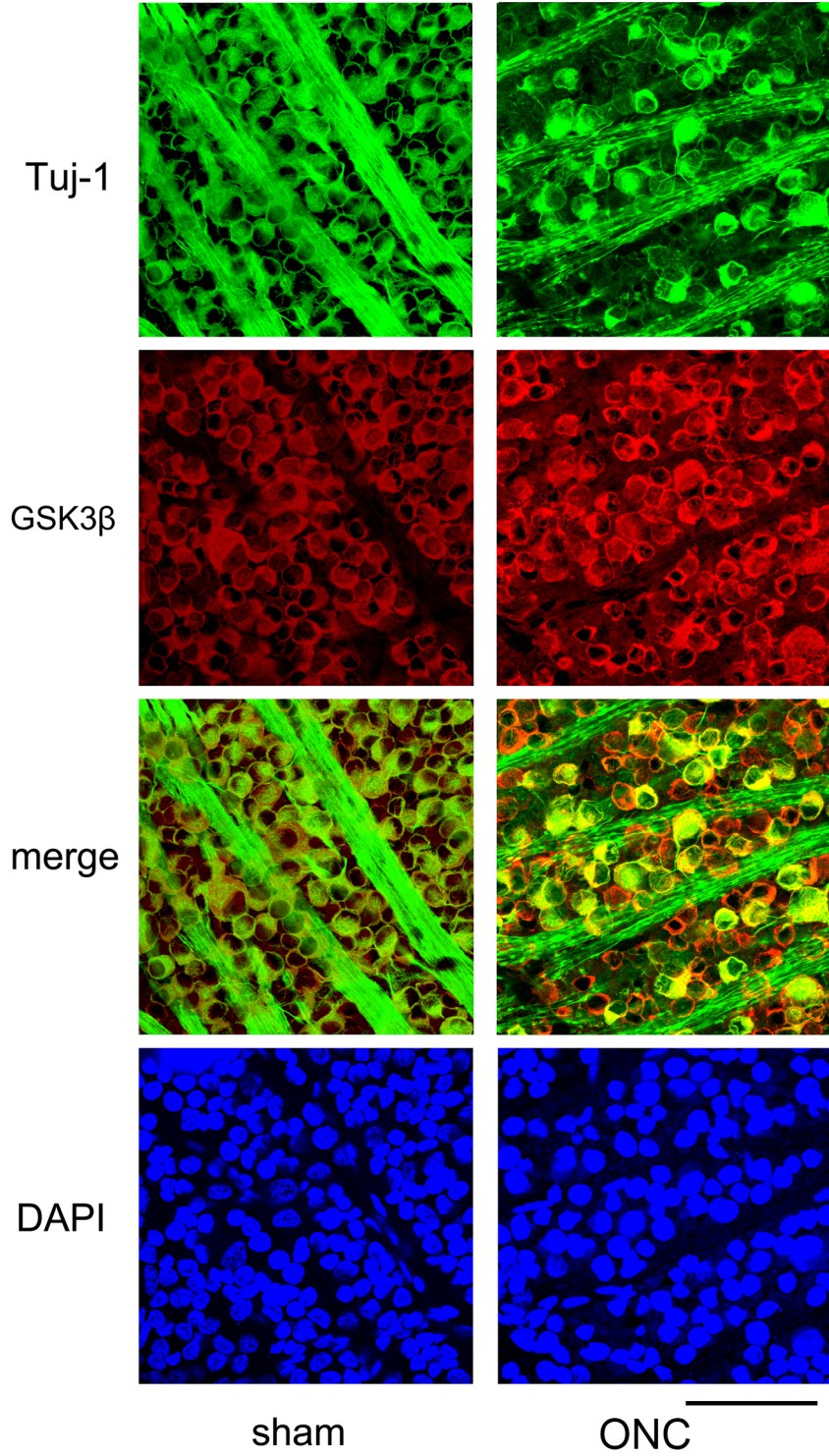

**Figure 7.** Additional confocal microscopic images of flat-mounted retinas co-labeled with Tuj-1 and GSK3β proteins are provided to complement our findings. In comparison to sham control (sham), Tuj-1-positivity (green) was lost in some cells on day 7 after ONC, but GSK3β (red) was still present in these cells (ONC). Nuclei were stained with DAPI (blue) (Bar = 100 μm).

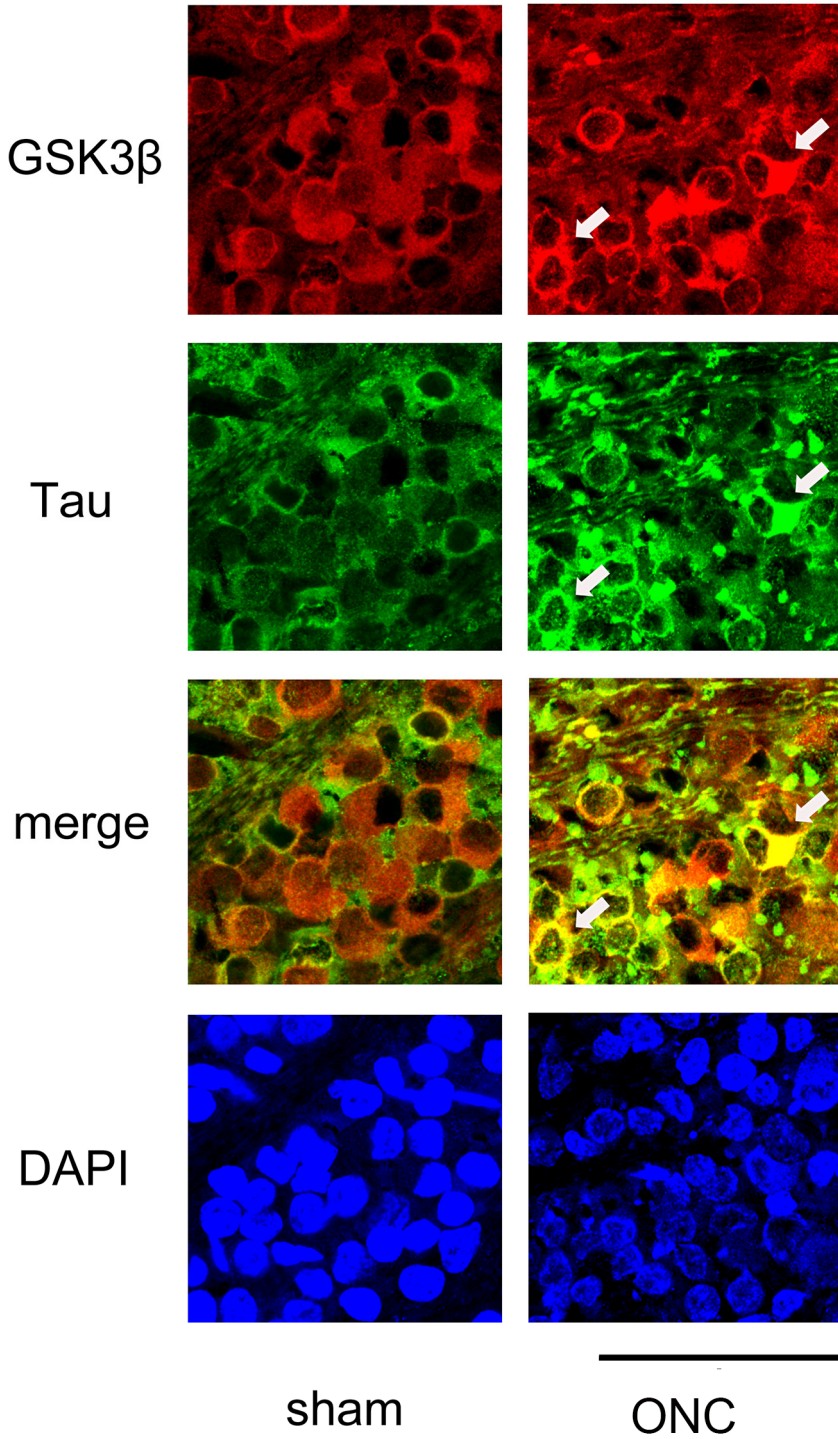

**Figure 8.** Confocal pictures of flat-mounted retinas co-labeled with antibodies against GSK3β and tau proteins for representative examples. GSK3β (red) was detected in the cell bodies of RGCs, and the expression of tau (green) was nearly uniform in the sham control group. In the ONC group, immune reactivity to tau (green) was increased in some RGCs that exhibited high levels of GSK3β (red) expression (arrows). Nuclei were stained with DAPI (blue). Retinas were obtained on day 7. (Bar = 100 μm).

### 3.4. ELISA for GSK3β

We lastly examined whether LiCl limited the amount of GSK3β on day 7 after ONC by an ELISA. The results are shown as fold changes to the sham control level (mean ± SD) in Figure 9. The levels of GSK3β in the placebo group increased to 2.0 ± 0.6-fold from the sham control levels ($p = 0.01$, Scheffe). Treatment with LiCl depressed the increase by 1.2 ± 0.2-fold ($p = 0.04$, Scheffe).

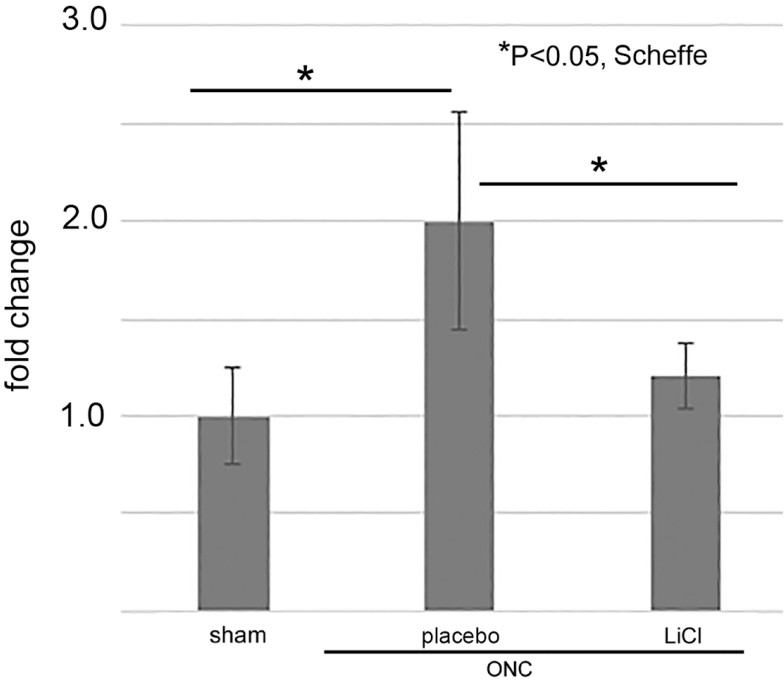

**Figure 9.** The levels of GSK3β were determined by ELISA and expressed by fold-changes to the sham control levels (mean ± SD). After ONC, the levels increased in the placebo group, but LiCl treatment depressed the increase ($n = 4$ per group with biological replicate).

## 4. Discussion

Consistent with our previous study [4], phosphorylated tau increased in the retina after ONC. Immuno-reactivity to GSK3β was semi-quantitatively measured and the level in RGCs increased over time after ONC. In addition, the ELISA showed that there was a 2.0-fold increase of GSK3β in the retina (in comparison to the sham control) on day 7. The expression of GSK3β was present in the somas of the RGCs and was intensified and co-localized with the expression of tau. Additionally, inhibition of GSK3β by LiCl resulted in decreased amounts of phosphorylated tau and a reduction in the degree of RGC loss on day 7. These results indicate optic nerve injuries may cause upregulation of GSK3β in the RGCs which may contribute to tauopathy-related death of RGCs.

Previously we demonstrated that reducing *tau* gene expression resulted in decreased levels of phosphorylated tau in the retina and rescued RGCs after ONC [4]. Temporal changes in tau expression in RGCs after ONC were determined by IHC, and tau immunoreactivity intensified over time during observation period on days 1, 3, and 7 post injury [4].

The relationship between tau pathology and neurotoxicity is complex and has not been fully clarified yet. However, it has been proposed that hyperphosphorylation of tau at specific sites can lead to the development of tau oligomers, which are associated with impaired binding to microtubules and axonal transport impairment [22]. Additionally, tau oligomers can cause excitotoxicity in RGCs [23], which may explain the toxic effects of increased phosphorylated tau on RGCs and subsequent decrease in the number of Tuj-1-stained RGCs on day 7.

Formation of tau oligomers involves hyper-phosphorylation of tau. We observed an increase of dimers of phosphorylated tau after ONC in this study. The results of pathological examination of the brain in Alzheimer's disease suggest that two proteins, cyclin-dependent kinase 5 (Cdk5) [5] and GSK3β [6], may be involved in the hyper-phosphorylation of tau. In addition, it has been reported that tauopathy-related changes occur in the diabetic retina, with GSK3β playing a crucial role [24].

Glycogen synthase kinase 3 (GSK3) is a type of serine/threonine protein kinase that plays a key role in regulating various cellular processes, including proliferation, differentiation, and adhesion [25]. Two isoforms, GSK3α and GSK3β, have been identified and both are ubiquitously expressed in the brain and GSK3β is increased in the post-mortem brain of Alzheimer's disease patients [26]. In addition, studies using IHC showed that GSK3β and phosphorylated tau are co-localized in the hippocampal neurons in Alzheimer's disease [26]. These results indicate that GSK3β is one of the main kinases involved in pathologic tau phosphorylation. Although the antibody against tau used in the present IHC is not specific to phosphorylated tau, WES assays suggested that the increase of retinal tau chiefly occurred in the phosphorylated isoform. Thus, our data that the expression of GSK3β was intensified throughout the observation period (days 1, 3 and 7 after ONC) with a reduction of Tuj-1 stained RGCs and co-localized with the expression of tau also suggest that GSK3β may be associated with tau phosphorylation in the RGCs after ONC because of its time-dependent increase and co-localization with tau in the RGCs.

GSK3β is the primary kinase that causes hyper-phosphorylation of tau species in mouse models of Alzheimer's disease, because the overexpression of GSK3β in mouse hippocampal neurons increases hyper-phosphorylated tau and causes neuronal death and memory impairment [27]. Conversely, reducing the kinase activity or the deletion of tau in GSK3β overexpressing mice leads to the recovery of learning abilities [28].

Activities of GSK3α and GSK3β are regulated by Akt-induced phosphorylation at their specific sites (GSK-3α at serine 21, GSK-3β at serine 9) [29]. It has been shown that ONC causes the rapid inactivation of phosphatidyl inositol-3-kinase (PI3K)/Akt system, which is associated with apoptosis of RGCs after ONC [30]. Thus, it is conceivable that there is a strong tendency for the increased GSK3β seen in our study to be activated by a reduction of Akt signaling pathway.

Previously, we showed that Cdk5 activation was associated with pathological phosphorylation of tau after ONC in rats [7]. We showed that roscovitine, a CDK5 inhibitor, reduced the levels of phosphorylated tau and the degree of loss of RGCs after ONC in rats. It has been shown that abnormal phosphorylation of tau species requires primary phosphorylation by Cdk-5 prior to the sequential action of GSK3β [31]. In addition, several studies have suggested cross-linkage between Cdk5 and GSK3β in the pathological phosphorylation of tau. For example, p25 (a Cdk5 activator) also binds to GSK3β and increases its kinase activities [32]. Calpain cleaves P35 to P25 and activates Cdk5, and GSK3β is also activated by calpain-induced cleavage of GSK3β [33]. Indeed, we previously found that the calpain signaling pathway was activated and associated with Cdk5 activation after ONC [7]. Based on the underlying data obtained from previous studies, including our own, we speculate that these two kinases may interact and play a role in the pathological phosphorylation observed in RGCs after ONC.

The present study was associated with several limitations. One limitation of our study was that we did not investigate the mechanisms by which Cdk5 and GSK3β are interconnected and promote the abnormal phosphorylation of tau. For example, inactivating one kinase could potentially affect the activity of the other kinase. Understanding the relationship between these kinases and their interplay could provide important insights into the mechanisms underlying tauopathy-related optic nerve damage. Another limitation was that we did not determine the changes of toxic tau oligomers in the retina by recently developed specific antibody. These issues need to be investigated further.

In conclusion, GSK3β was increased in the retina after ONC and its inhibition by LiCl reduced the amounts of retinal phosphorylated tau and depressed the loss of RGCs after

ONC. These findings indicate GSK3β may be a therapeutic target that may regulate optic nerve damage from increased tau levels in some optic nerve injuries.

**Author Contributions:** Y.F. and H.O. drafted this manuscript, collected the data, and reviewed the literature. T.H. helped with the experiments. S.T. and T.K. critically reviewed the final version of the manuscript. All authors have read and agreed to the published version of the manuscript.

**Funding:** This research was funded by Grant-in-Aid for Scientific Research (KAKENHI) (Grant No.22K09845) from the Japan Society for the Promotion of Science (Tokyo, Japan), and contract research grant (AS2020A000029493) from J&J Surgical Vision (AMO, Tokyo, Japan). The sponsors had no role in the design or conduct of this research.

**Institutional Review Board Statement:** The animal study protocol was approved by the Animal Use and Care Committee of Osaka Medical and Pharmaceutical University (No. 21025-A, approved on 25 February 2022).

**Informed Consent Statement:** Not applicable.

**Data Availability Statement:** The data that support the findings of this study are available on request from the corresponding author, Hidehiro Oku (hidehirooku@aol.com).

**Conflicts of Interest:** All authors certify that they have no affiliations with or involvement in any organization or entity with any financial interest (such as honoraria; educational grants; participation in speakers' bureaus; membership, employment, consultancies, stock ownership, or other equity interest; and expert testimony or patent-licensing arrangements), or non-financial interest (such as personal or professional relationships, affiliations, knowledge, or beliefs) in the subject matter or materials discussed in this manuscript.

## Abbreviations

| | |
|---|---|
| CNS | central nervous system |
| IHC | immunohistochemistry |
| RGCs | retinal ganglion cells |
| Cdk | cyclin-dependent kinase |
| GSK3β | glycogen synthase kinase 3β |
| ONC | optic nerve crush |
| LiCl | lithium chloride |
| Da | dalton |
| AIU | arbitrary intensity unit |
| ELISA | enzyme-linked immunosorbent assay |
| SDS | sodium dodecyl sulfate |
| ANOVA | analysis of variance |
| FBS | fetal bovine serum |
| BSA | bovine serum albumin |
| EDTA | ethylenediaminetetraacetic acid |
| PVDF | Polyvinylidene difluoride |
| PBS | Phosphate-buffered saline |
| T-PER | tissue protein extraction reagent |

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
