# Peer review of "Involvement of Glycogen Synthase Kinase 3β (GSK3β) in Formation of Phosphorylated Tau and Death of Retinal Ganglion Cells of Rats Caused by Optic Nerve Crush"

_cimb, doi:10.3390/cimb45090438_

Round 1

Reviewer 1 Report

Comments and Suggestions for Authors

The present article, entitled “Involvement of Glycogen Synthase Kinase 3β (GSK3β) in Formation of Phosphorylated Tau and Death of Retinal Ganglion Cells of Rats Caused by Optic Nerve Crush” by Fukiyama et al. investigates the consequences of Optic Nerve Crush in the retina of male rats by molecular and immunohistochemical techniques. Although investigating neuronal damage (through Tuj-1 expression) and the involvement of tau and expression GSK3β within the retina 3 and 5 days after ONC, the manuscript requires a reorganization to improve the clarity and relevancy of the results achieved.

Authors may consider re-organising the abstract to make the message clearer.

Regarding Material & Methods,

Please consider including some more details, and supporting literature, on the anaesthesia and euthanasia applied. Include also the way in which euthanasia was confirmed.

The Optic Nerve Crush (PNC) protocol is performed on the left eye, however it is not clear which retina is employed in the next experiment: Immunohistochemistry, and protein extraction and immunoblotting (as examples). For protein extraction, it is indicated that “at the designated time points…” although these time point are not clearly indicated at this point. Authors indicate that “an ophthalmoscopic examination was conducted to verify the absence of retinal ischemia and to confirm that the HIF-1α gene was unchanged, as determined by real-time PCR.”; however, these data are not reported for the present experiment. Please explain. The use of the same side retina (left), and not the contralateral, is a crucial point to consider the importance and relevance of the present findings.   

Authors should indicate how were the retina preparations “cover-slipped the following morning”.

To better understand how surviving RGCs were counted, authors may consider including a scheme or draw of the retinal flat mounts and the four areas considered. Why did authors use Tuj-1 as a neuronal marker instead of more precise marker for a RGCs, such as Brn3a? Please explain and present pros and cons.

Authors are suggested to include an experimental time-line to indicate the assays performed at the time points of interest: on days 1, 3 and 7 post-ONC. Please include the time points of the data measured in the Figure legend.

Authors reported that “the retinas were carefully extracted from the eyes using the method detailed by Winkler [15].” However, this citation, referring to the performance of an electroretinogram in the isolated rat retina, is quite old (1972); we may suggest using a more recent citation, mainly to be able to access the research work online, and thus improving the reproducibility and replicability of the results.

Regarding the Results section:

“Placebo” group is hardly used in animal models since no placebo effect can be evaluated in animals. Moreover, this “placebo” group [ONC + PBS intravitreal injection] should be considered as vehicle injected, and results should be similar to the “ONC alone” group. Authors should make an effort to explain why did they achieve different results when evaluating the “placebo” and the “ONC alone” group.

Authors indicate that “Immunoblotting was performed using pooled samples and measurements were performed in triplicate (n=3)”. However, is should be further clarified in the results section, and the number of retinas employed in each pool should be indicated. Figures 1 and 2 might be forged together, since results from western blots are presented as histograms in Figure 2. Results from western blots (presented in Figure 1 and quantified in Figure 2) require a more in depth explanation since “ONC alone” and “placebo” groups should be similar. However, it is not the case, since tau (>150kDa) and p-tau (>50kDa, around 100kDa and >150kDa) blots are more marked in the “placebo” that in the “ONC alone” group (more clear for the p-tau results). Moreover, authors may justify why only results from the monomer are presented, and not those from dimers or trimers. Actually, authors include it as a limitation: “we did not determine the changes of tau oligomer in the retina” although data from the western blots analysis seem to be available.

The investigation of the LiCl effects should include a different statistical analysis since two independent factors are considered in the experiment: the ONC (or sham surgery) and the drug, LiCl (or the vehicle). Therefore, a two way ANOVA has to be performed whenever the 4 groups are considered. Moreover, authors should not change the experimental groups names (ONC group into crush) to avoid misunderstandings.

Results from immunostaining present in Figure 3 seem to be confusing since images from the ONC-LiCL group appear unfocused, and the staining, both for Tuj-1 and Tau, seem to be diffuse, and not properly associated to fibres.

In Figure 4B, the units for the cell viability need to be included in the Y-axis more clearly; authors should include the ImageJ tool employed for the quantification of these immunohistochemically data. Authors indicate that “Tuj-1 staining in an area of 0.2 × 0.2 mm at the center of each image were counted.” please indicate how this process was done, and consider including a diagram.

Authors may re-organize and clarify the main results obtained in this study, including a graphical abstract showing the main changes observed following ONC, 3 and 7 days after the crush. Moreover, a discussion for the temporal changes following ONC should be included: 3 days, increase in p-tau, and discussion on the occurrence of polymerization, and/or the functionality of the protein tau itself; 7 days, decrease in cell viability (Tuj-1 staining), although images from Figure 6 are not clear enough), how did Tau expression change? Data from Figure 3 are not clear since Tau expression seem to be more diffuse, less intense, and quite widespread; and how was GSK3 β modified? It would have been of great interest to include immunohistochemical analyses of GSK3 β expression following LiCl in order to further to demonstrate its actions on retinal GSK3β.

Author Response

Thank you for your time to assess our manuscript.

We have revised our manuscript according to your instructions and comments as thoroughly as possible. We appreciate your constructive feedback on our manuscript. Due to time constraints before resubmission, we were unable to address all comments completely. Our responses are provided in detail below.

The present article, entitled “Involvement of Glycogen Synthase Kinase 3β (GSK3β) in Formation of Phosphorylated Tau and Death of Retinal Ganglion Cells of Rats Caused by Optic Nerve Crush” by Fukiyama et al. investigates the consequences of Optic Nerve Crush in the retina of male rats by molecular and immunohistochemical techniques. Although investigating neuronal damage (through Tuj-1 expression) and the involvement of tau and expression GSK3β within the retina 3 and 5 days after ONC, the manuscript requires a reorganization to improve the clarity and relevancy of the results achieved.

Authors may consider re-organising the abstract to make the message clearer.

Response.

Thank you for your suggestions. We have revised the Abstract to make importance of our findings clearer. In addition, we have created Graphic Abstract.

Regarding Material & Methods,

Please consider including some more details, and supporting literature, on the anaesthesia and euthanasia applied. Include also the way in which euthanasia was confirmed.

Response.

We have cited methods for anesthesia and euthanasia (line 102-109)

The Optic Nerve Crush (PNC) protocol is performed on the left eye, however it is not clear which retina is employed in the next experiment: Immunohistochemistry, and protein extraction and immunoblotting (as examples). For protein extraction, it is indicated that “at the designated time points…” although these time point are not clearly indicated at this point. Authors indicate that “an ophthalmoscopic examination was conducted to verify the absence of retinal ischemia and to confirm that the HIF-1α gene was unchanged, as determined by real-time PCR.”; however, these data are not reported for the present experiment. Please explain. The use of the same side retina (left), and not the contralateral, is a crucial point to consider the importance and relevance of the present findings. 

Response.

We are sorry for confusing you. We determined our methods did not cause retinal ischemia by confirming HIF 1α was stable in the previous study, but not in this study. We have properly corrected the description not to cause confusion. We used only left eyes that underwent ONC or sham operation. We did not use right eyes as controls because it has been shown that unilateral optic nerve injuries can affect gene expression in the contralateral retinas. Temporal changes in the expression of GSK3β using IHC were determined on days 1, 3 and 7 days after ONC. Neuroprotection by LiCl was determined on day 7. At these time points, animals were euthanized. In addition, protein extraction for immunoblot was performed on day 3 and for ELISA on day 7. Experimental timeline is now shown in Figure 1 in the revised manuscript.

Authors should indicate how were the retina preparations “cover-slipped the following morning”.

Response.

We have added the methodological description in the Lines 138-142.

To better understand how surviving RGCs were counted, authors may consider including a scheme or draw of the retinal flat mounts and the four areas considered. Why did authors use Tuj-1 as a neuronal marker instead of more precise marker for a RGCs, such as Brn3a? Please explain and present pros and cons.

Response.

We have shown a schematic diagram to count RGCs in Figure 1B.

We have not experienced to use Brn3a, but Tuj-1 is also widely used as a marker for retinal ganglion cells and is known from experience to label them clearly.

Authors are suggested to include an experimental time-line to indicate the assays performed at the time points of interest: on days 1, 3 and 7 post-ONC. Please include the time points of the data measured in the Figure legend.

Response.

In addition to the experimental timeline in Figure 1, we have described time points of assays in the Figure legends in the revised manuscript.

Authors reported that “the retinas were carefully extracted from the eyes using the method detailed by Winkler [15].” However, this citation, referring to the performance of an electroretinogram in the isolated rat retina, is quite old (1972); we may suggest using a more recent citation, mainly to be able to access the research work online, and thus improving the reproducibility and replicability of the results.

Response.

We have deleted the citation and described our methods in more detail (Line 131-132).

Regarding the Results section:

“Placebo” group is hardly used in animal models since no placebo effect can be evaluated in animals. Moreover, this “placebo” group [ONC + PBS intravitreal injection] should be considered as vehicle injected, and results should be similar to the “ONC alone” group. Authors should make an effort to explain why did they achieve different results when evaluating the “placebo” and the “ONC alone” group.

Response.

We aimed to determine whether vehicle injection into the vitreous itself has any effects on decreasing P-tau formation. In this context, vehicle injection did not decrease P-tau levels in the placebo group compared to the ONC alone group. Thus, LiCl itself has decreasing effects on P-tau. Regarding discrepancies between the ONC alone and placebo group, we have described possible explanations in page 6.

Authors indicate that “Immunoblotting was performed using pooled samples and measurements were performed in triplicate (n=3)”. However, is should be further clarified in the results section, and the number of retinas employed in each pool should be indicated. Figures 1 and 2 might be forged together, since results from western blots are presented as histograms in Figure 2. Results from western blots (presented in Figure 1 and quantified in Figure 2) require a more in depth explanation since “ONC alone” and “placebo” groups should be similar. However, it is not the case, since tau (>150kDa) and p-tau (>50kDa, around 100kDa and >150kDa) blots are more marked in the “placebo” that in the “ONC alone” group (more clear for the p-tau results). Moreover, authors may justify why only results from the monomer are presented, and not those from dimers or trimers. Actually, authors include it as a limitation: “we did not determine the changes of tau oligomer in the retina” although data from the western blots analysis seem to be available.

Response.

Thank you for your suggestions. Figures 1 and 2 are combined in the revised manuscript. In addition, we have determined P tau levels around 100 kDa bands which may represent dimers. Findings of immunoblots are described in more detail in page 6 and the Figure legends including discrepancies between the effects of ONC alone and placebo group.  

The investigation of the LiCl effects should include a different statistical analysis since two independent factors are considered in the experiment: the ONC (or sham surgery) and the drug, LiCl (or the vehicle). Therefore, a two way ANOVA has to be performed whenever the 4 groups are considered. Moreover, authors should not change the experimental groups names (ONC group into crush) to avoid misunderstandings.

Response.

I understand that one-way analysis of variance (ANOVA) is used to investigate whether the difference in mean values between groups is statistically significant. It can be used to compare whether there is a statistically significant difference between the control group (sham control in this study) and the experimental groups (ONC groups). Scheffe test was used to compare between individual groups in the revised manuscript. We have changed labelling of “crush” to “ONC”.

Results from immunostaining present in Figure 3 seem to be confusing since images from the ONC-LiCL group appear unfocused, and the staining, both for Tuj-1 and Tau, seem to be diffuse, and not properly associated to fibres.

Response.

Thank you for your suggestion, and we have created new version of Figure3. Images of LiCl group are replaced by other images.

In Figure 4B, the units for the cell viability need to be included in the Y-axis more clearly; authors should include the ImageJ tool employed for the quantification of these immunohistochemically data. Authors indicate that “Tuj-1 staining in an area of 0.2 × 0.2 mm at the center of each image were counted.” please indicate how this process was done, and consider including a diagram.

Response.

The units of Y-axis in the Figure 4 have been made clear. We have verified it by viewing with higher magnification. We have shown a schematic diagram to count RGCs in Figure 1B.

Authors may re-organize and clarify the main results obtained in this study, including a graphical abstract showing the main changes observed following ONC, 3 and 7 days after the crush. Moreover, a discussion for the temporal changes following ONC should be included: 3 days, increase in p-tau, and discussion on the occurrence of polymerization, and/or the functionality of the protein tau itself; 7 days, decrease in cell viability (Tuj-1 staining), although images from Figure 6 are not clear enough), how did Tau expression change? Data from Figure 3 are not clear since Tau expression seem to be more diffuse, less intense, and quite widespread; and how was GSK3 β modified? It would have been of great interest to include immunohistochemical analyses of GSK3 β expression following LiCl in order to further to demonstrate its actions on retinal GSK3β.

Response.

The Discussion section is extensively revised to explain roles of GSK3β on tauopathy. Temporal changes in the expression of tau in the RGCs were done in our previous article and tau was increased overtime. The findings were cited in the Discussion (2nd paragraph in the Discussion).

Reviewer 2 Report

Comments and Suggestions for Authors

Fukiyama et al

Involvement of glycogen synthase kinase 3b in formation of phosphorylated tau and death of retinal ganglion cells of rats caused by optic nerve crush

Fukiyama et al showed convincingly that GSK-3b elevation is one of the key molecular events associated with loss of retinal ganglion cells after optic nerve crush. That the LiCl treatment alleviates GSK-3b upregulation and the loss of retinal ganglion cells affords a piece of critical evidence for the damning effect resulting from this kinase. However, the involvement of tau phosphorylation is less convincing. There are many phosphorylated tau antibodies readily available. The authors only used one such reagent in a single immunoblot. No IHC was done with phosphorylated tau antibodies. Furthermore, the link between the general tau Ab and the phos-Ser396 is weak. Therefore, the bands seen in Figure 1 are not at all confirmatory. IP-Western is needed to verify the identity of these “tau” bands.

Figure 2: n of 3 technical replica is not sufficient. Biological repeats are needed. Besides, it was not explained as to why the sham control showed such a high level of p-tau. How can one calculate the P value from sham against sham? This figure is very weak.

Minor concern: The authors need to update their knowledge in tau biology and function. Much of what has been stated in the Introduction is unfounded. The physiological function of tau is unknown. Insoluble tau is unlikely to be neurotoxic or diffusible.

“[C]ross-linkage between CDK5 and GSK-3b” (line 360) is obscure.

Author Response

Thank you for your time to assess our manuscript. 

We have revised our manuscript according to your instructions and comments.  Due to time constraints before resubmission, we were unable to address all the comments completely. Our responses are provided below.

Fukiyama et al showed convincingly that GSK-3b elevation is one of the key molecular events associated with loss of retinal ganglion cells after optic nerve crush. That the LiCl treatment alleviates GSK-3b upregulation and the loss of retinal ganglion cells affords a piece of critical evidence for the damning effect resulting from this kinase. However, the involvement of tau phosphorylation is less convincing. There are many phosphorylated tau antibodies readily available. The authors only used one such reagent in a single immunoblot. No IHC was done with phosphorylated tau antibodies. Furthermore, the link between the general tau Ab and the phos-Ser396 is weak. Therefore, the bands seen in Figure 1 are not at all confirmatory. IP-Western is needed to verify the identity of these “tau” bands.

Figure 2: n of 3 technical replica is not sufficient. Biological repeats are needed. Besides, it was not explained as to why the sham control showed such a high level of p-tau. How can one calculate the P value from sham against sham? This figure is very weak.

Response.

Thank you for your suggestions and we agree with you in that biological replicate is much more suitable.  In our experimental experience using rats, 3 retinas are necessary to extract enough proteins for immunoblot. Biological replicate is difficult to perform because much more rats must be sacrificed. We recognize that phosphorylation of tau occurs during physiological process. We used Scheffe test to determine significance in the revised manuscript.

Minor concern: The authors need to update their knowledge in tau biology and function. Much of what has been stated in the Introduction is unfounded. The physiological function of tau is unknown. Insoluble tau is unlikely to be neurotoxic or diffusible.

Response.

We have described your concern in the Discussion.

The relationship between tau pathology and neurotoxicity is complex and has not been fully clarified yet. (3rd paragraph in the Discussion)

Reviewer 3 Report

Comments and Suggestions for Authors

The manuscript of Fukiyama et al confirms the link between GSK3β and tau phosphorylation in the RGCs after Optic Nerve Crush because of its time-dependent upsurge and co-localization with tau in the RGCs. The inhibition of GSK3β by LiCl reduced the amounts of retinal phosphorylated tau and decreased the damage of RGCs after ONC. The Title and Abstract are illustrative of the study. The summary and introduction give a clear review of the need of the study, methods, results, and conclusions within the word limit. The research work is very intriguing; the manuscript is well organized and clearly written. The figures are clear and are strong and informative to support the conclusion. In future, the authors should consider studying the molecular mechanism between Cdk5 and GSK3β at both Transcription and Translational levels. However, the following comments may be addressed before acceptance:

1.      Novelty of the manuscript must be better emphasized

2.      What is the reason behind the selection of Wistar rats, may be added? 

3.      Line 70: Please consider adding the number of rats used in the study.

4.      The western blot in Fig 1 does not clearly correlate with the Fold change (densitometry) in Figure 2.

5.      Statistical Analysis may be reverified.

Author Response

Thank you for your time to assess our manuscript.

We have revised our manuscript according to your instructions and comments. Our responses to are provided in detail below.

The manuscript of Fukiyama et al confirms the link between GSK3β and tau phosphorylation in the RGCs after Optic Nerve Crush because of its time-dependent upsurge and co-localization with tau in the RGCs. The inhibition of GSK3β by LiCl reduced the amounts of retinal phosphorylated tau and decreased the damage of RGCs after ONC. The Title and Abstract are illustrative of the study. The summary and introduction give a clear review of the need of the study, methods, results, and conclusions within the word limit. The research work is very intriguing; the manuscript is well organized and clearly written. The figures are clear and are strong and informative to support the conclusion. In future, the authors should consider studying the molecular mechanism between Cdk5 and GSK3β at both Transcription and Translational levels. However, the following comments may be addressed before acceptance:

  1. Novelty of the manuscript must be better emphasized.

Response.

We have added Graphic Abstract. In the first paragraph of the Discussion, we have added the following descriptions.

These results indicate optic nerve injuries may cause upregulation of GSK3β in the RGCs which may contribute to tauopathy-related death of RGCs.

  1. What is the reason behind the selection of Wistar rats, may be added?

Response.

This is our opinion from the experience, but Wistar rats have less pigmentation and are suitable for fluorescence immunostaining of the retina.

  1. Line 70: Please consider adding the number of rats used in the study.

Response.

We have described the number of animals used to obtain the results shown in the manuscript. (2.1. Animals, page 2)

  1. The western blot in Fig 1 does not clearly correlate with the Fold change (densitometry) in Figure 2.
  2. Statistical Analysis may be reverified.

Response.

We have recalculated the densities of the immunoblots, and the data are somewhat different from the previous results. We have also measured levels of the dimers, and the data are included in the revised manuscript. Furthermore, we have used Scheffe's test in the revised manuscript.

Round 2

Reviewer 2 Report

Comments and Suggestions for Authors

The manuscript in its current form is scientifically identical to the previous version. None of the major concerns was addressed. The bands designated as tau remain unresolved experimentally. There still are no biological repeats for the animal work. Tau, in the Introduction, still is claimed to cause neuronal death by tangles. 

Author Response

Thank you for your instruction about tau pathology. We appreciate your pointing out our misunderstanding about tau pathology. We have revised 1st paragraph in the Introduction section of our manuscript.

For the immunoblot analyses, we initially confirmed the presence of bands at the expected molecular weight of the P-tau (ser396) antibody using pooled samples made up of equal volumes of samples from all experimental groups combined. The results indicated that bands appeared around the predicted molecular weight of the antibody (50-60 kDa). Additionally, bands with a higher molecular weight around 100 kDa were also clearly visible. No significant differences were observed between tests using the same samples (below, preliminary experiments). Therefore, it is reasonable to conclude that the bands around 50-60 and 100 kDa represent monomers and dimers of phosphorylated tau, respectively.

Despite slight variations in the loading conditions, bands were also observed around 50-60 kDa and 100 kDa and significant differences in the expression were observed between the experimental groups (Figure 2B). Thus, it is reasonable to assume that these differences are due to the experimental conditions.

We recognize that biological replicates are valuable, but using them would require sacrificing a greater number of rats. Given the current 3R policy on animal welfare, the use of more animals is considered to be undesirable. Technical replicates can also be important, as they demonstrate the reliability of the measurements.

Round 3

Reviewer 2 Report

Comments and Suggestions for Authors

1. The attempt by the authors to use animal welfare to avert the need of rigor and reproducibility does not meet the scientific standard and, ironically, wasted the lives of those animals already sacrificed. A minimal number of rats for biological repeats of the most essential experiment is expected.

2. The identity of tau and phosphorylated tau needs to be verified by IP-western. The discussion provided by the authors does not explain the vastly different patterns of the the immunoblots. 

Author Response

To the Editor,

We have revised our manuscript according to the comments of the reviewers 2.

We assessed alterations in the expression of phosphorylated (ser 396) and total tau utilizing Simple Western (WES) assays, thereby reducing the number of animals required for sacrifice. This system requires a considerably smaller protein amount for the assay, with one retina sufficient for a single assay. Employing this approach, we conducted immunoassays with a sample size of n = 3 in each group, representing biological replicates.

Various isoforms of tau are recognized to exist, and the antibody employed in this study detects multiple bands within the range of 50-60 kDa (as illustrated in the image below from Santa Cruz). We identified and quantified three distinct bands falling within the 50-60 kDa spectrum to determine the overall amount.

We did not perform IP western that the reviewer 2 told us to do. We think it is not necessary for the purpose of this study.

Thank you for you time to evaluate our manuscript. Uncropped images for immunoassays will appear in the following pages.

Sincerely,

Hidehiro Oku, MD., PhD.

Professor, Department of Ophthalmology,

Osaka Medical Pharmaceutical University
